# Lower Blood Vitamin D Levels Are Associated with Depressive Symptoms in a Population of Older Adults in Kuwait: A Cross-Sectional Study

**DOI:** 10.3390/nu14081548

**Published:** 2022-04-08

**Authors:** Thurayya Albolushi, Manal Bouhaimed, Jeremey Spencer

**Affiliations:** 1Hugh Sinclair Unit of Human Nutrition, Department of Food and Nutritional Sciences, School of Chemistry, Food and Pharmacy University of Reading, Reading RG6 6AP, UK; j.p.e.spencer@reading.ac.uk; 2Department of Community Medicine and Behavioral Sciences, Faculty of Medicine, Kuwait University, P.O. Box 24923, Safat 13110, Kuwait; M.bouhaimed@ku.edu.kw

**Keywords:** 25 hydroxyvitamin D, depressive symptoms, cross-sectional study, mental health, aging

## Abstract

Low serum vitamin D has been associated with an increased risk of neuropsychiatry disorders. This study aimed to examine the association between vitamin D deficiency and depression in adults aged 65 years and older. This cross-sectional study was conducted in seven primary healthcare centers across Kuwait (November 2020 to June 2021). The participants (*n* = 237) had their serum vitamin D 25-(OH)-D concentrations (analyzed by LC-MS) classified as sufficient, ≥75 nmol/L (30 ng/mL); insufficient, 50–75 nmol/L (20–30 ng/mL); or deficient, <50 nmol/L (20 ng/mL). Depressive symptoms were evaluated using the 15-Item Geriatric Depression Scale (15-item GDS). The mean serum 25-OH-D levels (nmol/L) in volunteers with normal, mild, moderate, and severe depression were 100.0 ± 31.7, 71.2 ± 38.6, 58.6 ± 30.1 and 49.0 ± 6.93, respectively (*p* < 0.001). The participants in the vitamin D sufficiency group were significantly less likely to exhibit depressive symptoms (88.2%) than patients with mild (36%) and moderate (21%) depression (*p* < 0.001). Ordinal logistic regression showed that vitamin D deficiency (OR = 19.7, 95% CI 5.60, 74.86, *p* < 0.001) and insufficiency (OR = 6.40, 95% CI 2.20, 19.91, *p* < 0.001) were associated with higher odds of having depressive symptoms. A low serum vitamin D level is a significant predictor of symptoms of depression among older individuals.

## 1. Introduction

An estimated 322 million individuals suffer from depression worldwide, representing 4.4% of the global population. Depression is a leading cause of global disability [1] and is associated with reduced productivity, increased mortality, and high healthcare costs (both directly and indirectly) [2,3]. Moreover, depressive symptoms are common in the older adults and especially among the most senior adults [4]. The burden of depression later in life is expected to increase in Kuwait, as the proportion of its population who are aged 65 or over—currently at 4%—continues to grow at a rapid rate. This proportion is expected to increase to 4.41% of the overall population by mid-2025, and, by mid-2050, is expected to increase significantly to 17.90% [5]. Many adverse outcomes may be encountered by people who are older experiencing symptoms of depression, such as cognitive impairment, functional decline, lower quality of life, and disability [6]. While a full explanation of the pathophysiology and etiology of depression has yet to be provided, recent evidence indicates that various biological factors could impact mood [7]. Among these, vitamin D has emerged as a prominent nutrient associated with good mental health [8].

Vitamin D deficiency is a major public health problem worldwide, especially among older adults [9]. It has been linked to chronic diseases such as frailty, sarcopenia, osteoporosis, mild cognitive impairment, dementia, depression, and type 2 diabetes mellitus (T2DM) [10,11]. Older people are at particular risk of hypovitaminosis of vitamin D, potentially due to a number of lifestyle factors such as a lack of sun exposure, reduced diet quality, and age-related decline in the efficiency of vitamin D synthesis and metabolism [12].

The potential association between depression and vitamin D deficiency is supported by systematic reviews and meta-analyses of observational studies [13,14] as well as by other observational research evidence [15], where participants with severe vitamin D deficiency tend to be older and more likely to suffer from depressive symptoms [16,17,18,19,20]. However, other studies show no clear association or else mixed results [21,22,23]. The uncertainty of the relationship between depression and vitamin D may be further complicated by differences and limitations in methodological design. Most clinical research studies conducted in this area have used a limited sample size [24], administering symptom questionnaires to measure depression in general terms but not adopting methods such as clinical interviews to carry out psychiatric diagnoses, or have used the (less optimal) measurement of vitamin D via laboratory methods [4,23,25,26]. Moreover, most studies on this topic have been conducted among the residents of Western countries [18,25] or Asian populations [26].

This study aims to better understand the association between vitamin D status and depression among older adults in Kuwait while addressing many of the a forementioned limitations of the existing literature. In particular, it was hypothesized that a low plasma vitamin D < 50 nmol/L (20 ng/mL) [27] would predict symptoms of depression in older adults.

## 2. Materials and Methods

### 2.1. Population

The study was conducted in Kuwait in a population aged 65 years and over. A cross-sectional analysis was conducted between November 2020 and June 2021 across seven primary healthcare centers, which provide geriatric care through specialized geriatric clinics across Kuwait. A list of these primary healthcare centers was obtained from the Kuwait Ministry of Health (MoH). It included 18 geriatric clinics distributed across all six Governorates in the country. The seven healthcare centers were selected via stratified random sampling from these six Governates. The seven centers were selected as follows: one each from the Capital, Hawaly, Jahra, Mubarak Al-Kabeer, and Ahmadi and two from Farwanya. Therefore, this research could be described as nationally representative.

Participants were deemed eligible for the study if they were Kuwaiti citizens aged 65 years or above, with no global cognitive impairment (e.g., dementia and Alzheimer’s disease) and/or clinical diagnosis of depression, and not taking any anti-depression medication that could affect the outcome measures. The participants had to be Arabic speakers, with Arabic as their primary language, and had to have no vision or hearing problems that could prevent them from understanding the instructions and providing informed consent. Individuals with disabilities (e.g., bedridden), those reluctant to participate, and those with any disease that could affect vitamin D synthesis (e.g., liver disease, kidney disease, and skin cancer) were excluded from this study. Screening was performed using the measurement of Na, K, Cl, creatinine, albumin, urea, uric acid, and total bilirubin as well as other blood tests (gamma DT level, ALT, ALP, and CBC) to rule out poor kidney or liver function. Only participants with normal kidney function were recruited for the study. Moreover, only participants with complete data for depressive symptoms and measured 25-hydroxyvitamin D (25 [OH]D) were included in the analyses. This is summarized in the CONSORT recruitment diagram (see Figure 1).

To measure the association between variables (for example, between vitamin D and depression), the sample size was determined using the G-power program at the medium effect size of 0.3, a power of 0.80, and an alpha level of 0.05 to set the association [28]. After conducting the analysis with the power sample size, a minimum sample size was determined as 237 patients.

### 2.2. Ethical Approval and Informed Consent

This study was approved by the University of Reading Ethics Committee for Clinical Research (approval protocol no.: URCE 19/47) and by the Kuwait Ministry of Health Standing Committee for the Co-ordination of Medical Research (approval protocol no.: CB20-63MM-01). Participation then commenced with signing an informed consent form once the study’s aims, objectives, and collection procedures had been explained individually to each older adult invited. Those willing to participate signed the consent form, wherein the risks and benefits of the study were clearly explained. The study followed the principles of the Declaration of Helsinki.

### 2.3. Primary Outcome

Depression levels were measured using the Geriatric Depression Scale-15 (GDS-15) [29].

### 2.4. Data Collection and Covariates

The study covariates were measured based on variables in the extant literature [13,30], which could potentially confound symptoms of depression with serum 25(OH)D.

Sociodemographic and Lifestyle Factors, and Chronic Disease

Initial telephone screening interviews were carried out to identify potentially suitable participants. Research data were collected through one-on-one interviews, patient registries, and questionnaires. The data collected included the participants’ socioeconomic and demographic variables, such as their age, marital status (divorced, married, single, and widowed), educational level (no formal education, primary/intermediate, high school, diploma, university, or above), and income, and a lifestyle habits questionnaire was administered to ascertain smoking status (yes/no), alcohol use (yes/no), and sleep duration/patterns. Physical activity was measured with the Physical Activity Scale for the Elderly (PASE), specifically the International Physical Activity Questionnaire for the Elderly (IPAQ) [31]. In its officially validated, Arabic, short-version format, the IPAQ has been demonstrated to have acceptable validity and reliability when tested on a sample of Lebanese adults [32]. Patients’ health history was also obtained, such as any previous diagnosis of chronic diseases (e.g., type-2 diabetes, dyslipidemia, hypertension, cardiovascular disease (CVD), and osteoporosis). The researchers had access to patients’ medical history/records, and all participants were referred by medical practitioners who physically examined them to confirm previous diagnoses. Blood pressure was measured from the left upper arm using a mercury sphygmomanometer. All measurements were performed early in the morning prior to eating or taking any medications. Three separate blood pressure readings were taken and averaged. The physicians measuring blood pressure followed the International Society of Hypertension Global Hypertension Practice Guidelines [33]. Cognitive function was assessed by administering the 30-item Mini-Mental State Examination (MMSE), whereby scores lower than 24 would indicate cognitive impairment [23]. The MMSE scale has also been validated in an Arabic version [34].

Nutritional status was assessed according to anthropomorphic parameters (i.e., weight, height, body mass index (BMI), waist circumference, hip circumference, and waist–hip ratio (WHR)). BMI and WHR were classified according to the WHO cut-off points [35]. The Fitzpatrick Classification of Skin Phototype was applied by one assessor (for consistency) to measure the color of the participants’ skin as (I) very fair, (II) fair, (III) fair to medium, (IV) medium, (V) olive or dark, and (VI) very dark with deep pigmentation [36]. A sun exposure questionnaire was administered to assess the participants’ attitudes towards sun exposure (e.g., sun exposure time, sun cream use, and clothing worn) [37].

Information about vitamin D and calcium supplementation was collected using the Food Frequency Questionnaire (SFFQ), which was validated for the setting. This was a modified version of a previously validated questionnaire [38,39]. The original questionnaire was in English and was then translated into Arabic. From the questioner results, positive correlations were observed of daily vitamin D (r = 0.82, *p* < 0.001) and calcium intake (r = 0.74, *p* < 0.001). Food models or serving containers were used to estimate serving size. The SFFQ also included questions about the regular use of vitamin D and calcium supplements.

Lifestyle factors—for example, physical activity, smoking status, BMI, and diet—have been indicated as bidirectionally associated with depressive symptoms [40]. Therefore, lifestyle factors may be confounders in the relationship between symptoms of depression and vitamin D levels. It is therefore important for lifestyle factors to be considered in the statistical analysis. The entire questionnaire in this study was carefully developed and validated after an extensive review of the literature and pilot testing with 10 participants who were not included in the main study.

### 2.5. Biochemical Assessment

Assessment of Serum Vitamin D, Parathyroid Hormone (PTH), and the Biochemistry Test

A trained nurse collected 10 mL of blood from each participant and placed the sample in a tube containing gel (SST II Advance, BD Vacutainer). The samples were then shielded from the light. On the day of collection, the samples were centrifuged at 2000× *g* for 15 min; the serum was then poured into Eppendorf tubes and stored at −80 °C, until the samples were required for analysis [41]. Serum 25-(OH)-D concentration was applied as a measure of vitamin D level because this is the main circulating form of vitamin D. It comprises vitamin D consumed in food and vitamin D produced through sun exposure [41]. Qualified phlebotomists collected fasting blood samples, which were subsequently measured in a College of American Pathologists-accredited laboratory using liquid chromatography-tandem mass spectrometry (MC/MS/MS). This method is recommended for assessing vitamin D status in epidemiological studies [41,42].

In accordance with Endocrine Society guidelines, the following cut-offs for 25-(OH)-D were applied in this study to define vitamin D sufficiency, ≥75 nmol/L (30 ng/mL); vitamin D insufficiency, 50–75 nmol/L (20–30 ng/mL); and vitamin D deficiency, <50 nmol/L (20 ng/mL) [27]. Additionally, serum intact PTH was measured using the access intact PTH electrochemiluminescence immunoassay, with a commercial kit version of the Cobas E601 module analyzer, in accordance with the manufacturer’s instructions. The PTH reference range is 1.6–6.9 pmol/L when a patient is normocalcemic. Furthermore, serum cortisol was measured using electrochemiluminescence with e411/ELECSYS. In this procedure, serum cortisol samples were collected early in the morning (07.00–08.00). Additionally, fasting blood glucose (serum mmol/L) was measured with the Architect c4000 clinical chemistry analyzer, and serum calcium was measured using Arsenazo III dye (Architect plus c 4000, Abbott, Abbott Park, IL, USA). In addition, phosphorus (serum mmol/L) levels were evaluated with phosphomolybdate UV (Architect plus c 4000, Abbott, Abbott Park, IL, USA), as was alkaline phosphatase U/L.

### 2.6. Assessment of Depressive Symptoms

The participants’ depressive symptoms were clinically evaluated by a geriatrician and assessed using the GDS-15. This scale was developed by Yesavage et al. [29] to measure symptoms of depression in older populations. The GDS-15, which has been validated for use as a clinical instrument [32], can either be self-administered or applied as an interview guide. The advantage of this questionnaire is that the questions are in a simple yes/no format, making them easy for older subjects with impaired cognitive function to understand. For instance, the version used takes approximately 5–7 min to complete, rendering it suitable for respondents who may be easily fatigued or limited in their ability to concentrate for prolonged periods. The GDS-15 has been validated and translated into Arabic from English by a professional translator and two bilingual psychiatrists [43,44]. The Arabic version was adopted for this study. The scale has 15 dichotomous items, with a possible score of 0–15 [45]. A score of 0–4 indicates no depression, a score of 5–8 indicates mild depression, a score of 9–11 indicates moderate depression, and a score of 12–15 indicates severe depression [46]. For this study, the GDS-15 was applied as an interview guide. A researcher verbally asked the participants each question and recorded their responses.

### 2.7. Statistical Analysis

Frequencies and percentages were used to summarize categorical variables, whereas continuous variables were summarized using mean and standard deviation (SD) as the data were tested for normal distribution. Univariate analysis was performed with a Chi-square test of independence for categorical variables. One-way ANOVA was used to assess the association between the category of depression and continuous variables. Ordinal logistic regression (OLR) was used to assess the association between the category of 25-OH-D (deficiency, insufficiency, and sufficiency) and depression (normal, mild, moderate, and severe). The model was adjusted for gender, age, marital status, income, sun exposure, sleep duration (hours/day), activity, hypertension, cardiovascular disease, and dyslipidemia. The model was also adjusted for PTH concentration and vitamin D supplementation. The exponentiated coefficients for OLR represent the odds of having a higher depression score (y > j) compared with having a lower score (y ≤ j). Brant’s test was conducted to check the proportionality assumption. Brant’s test assesses whether the observed deviations from OLR are greater than those that would be expected through change alone. In addition, a likelihood ratio test was applied to compare the ordinal and multinomial logistic regression models. Post hoc pairwise comparisons of the level of depressive symptoms and vitamin D status were also conducted. *p*-values < 0.05 were considered statistically significant. Data analysis was performed using the SPSS software package, version 27 (SPSS Inc., Chicago, IL, USA) and R v 3.6.3.

## 3. Results

### 3.1. Characteristics of the Study Samples

A total of 237 participants were included in this study (54% female; 46% male) (Table 1). Various socioeconomic and demographic factors were associated with depressive symptoms (Table 1 and Table 2). For instance, gender, marital status, income, sleep duration, physical activity, walking, serum 25-(OH)-D level, vitamin D supplementation, calcium intake, cardiovascular disease, dyslipidemia, and sun exposure were associated with symptoms of depression. Similar numbers of participants were categorized with vitamin D sufficiency, insufficiency, and deficiency (36.7%, 32.9% and 30.4%, respectively). Most of the participants with deficiency and insufficiency showed mild and moderate depression symptoms, while most of the participants with sufficiency showed normal or mild depression symptoms (Table 2). A total of 100 participants were taking vitamin D supplements, of which 28 participants did not report any depressive symptoms, 43 had mild depression, and 29 had moderate depression (Table 1).

The GDS-15 categories were significantly associated with 25-(OH)-D, PO4 and PTH levels (Table 3). The mean 25(OH)D values for participants in the normal, mild, moderate, and severe categories were 100 ± 31.7, 71.2 ± 38.6, 58.6 ± 30.1 and 49.0 ± 6.93, respectively (*p* < 0.001). The mean PO4 values for participants with severe (1.43 ± 0.29) depression were higher than those for participants with no (1.13 ± 0.13), mild (1.12 ± 0.17), and moderate (1.12 ± 0.16) depressive symptoms. The mean PTH values in participants with severe (15.4 ± 18.6) depression were also higher than those in participants with normal (5.55 ± 2.17), mild (5.89 ± 2.45), and moderate (6.19 ± 3.23) depression. No significant difference was observed between participants with normal, mild, moderate, and severe depression in terms of other laboratory parameter factors—namely, ALP, Ca, cortisol (AM), blood glucose and insulin, BMI, or WHR. Waist and hip measurements were significantly different between the groups (*p* < 0.001).

#### 3.1.1. Association between Symptoms of Depression (GDS-15) and 25-(OH)-D Levels

Welch’s test was statistically significant (Fwelch (3, 15.12) = 20.18, *p* < 0.001, $E\left(\omega_{p}^{2}\right)$ = 0.75, CI (0.43, 0.86)), indicating an association between the GDS-15 scores and 25(OH)D levels. Post hoc pairwise comparisons showed that the average 25-(OH)-D levels (Appendix A) were significantly higher in normal participants than in participants with mild, moderate, or severe depressive symptoms (*p* < 0.001 for all pairwise comparisons).

#### 3.1.2. Association between GDS-15 and 25(OH)D Grouped by Variables

When the analysis was stratified by gender, a statistically significant association was observed between the GDS-15 scores and 25(OH)D levels among the female (Fwelch (3, 17.96) = 18.49, *p* < 0.001, $E\left(\omega_{p}^{2}\right)$ = 0.70, CI (0.39, 0.83)) and male (Fwelch (2, 39.33) = 12.93, *p* < 0.001, ${\hat{\omega }}_{p}^{2}$ = 0.36, CI [0.12, 0.54]) participants. Pairwise comparisons indicated a statistically significant difference in the average 25(OH)D values among the pairs: normal–moderate (*p* = 0.001), normal–severe (*p* < 0.001), and mild–severe (*p* = 0.002) in the female participants (Appendix A). Moreover, in the male participants, a significant difference was observed between the pairs: normal–mild (*p* = 0.002) and normal–moderate (*p* < 0.001).

A significant association was observed between the GDS-15 scores and average 25-OH-D levels in participants receiving vitamin D supplements (Fwelch (2, 3.46) = 22.28, *p* = 0.011, $E\left(\omega_{p}^{2}\right)$ = 0.87, CI (0.00, 0.96)). Post hoc pairwise comparisons indicated a significant difference in average GDS-15 scores between participants with insufficiency and sufficiency (*p* = 0.002). Only two participants who received supplementation displayed 25-OH-D deficiency.

Appendix A present the association between GDS-15 and 25-(OH)-D grouped by cardiovascular category and level of physical activity.

#### 3.1.3. Ordinal Logistic Regression (OLR) Analysis for the Association between Depressive Symptoms and 25-(OH)-D Status

The results (Figure 2) showed that 25-(OH)-D levels were significantly associated with depression after adjusting for other variables in the model. The odds of a higher depression score were significantly higher in participants with insufficient 25-(OH)-D than in participants with 25-OH-D sufficiency (OR = 6.40, *p* < 0.001). The odds were even lower in participants with sufficient 25-(OH)-D (OR = 19.70, *p* < 0.001). Gender displayed a statistically significant association with depression (OR = 0.34, *p* = 0.01), whereas age and marital status did not. Other factors that demonstrated a statistically significant association with depression were sleep duration in hours and physical activity. A longer sleep duration (hours/day) was associated with lower odds of having a high depression score (OR = 0.75, *p* = 0.03) (Appendix A). Similarly, physical activity was associated with lower odds of having a high depression score (OR = 0.34, *p* = 0.001). This indicates that the odds of having a high depression score were 66% lower in respondents who engaged in physical activity than in those who did not. Neither the presence of comorbidities nor monthly income showed a statistically significant association with the depression score.

## 4. Discussion

This study showed an association between the serum 25-OH-D concentration and depressive symptom scores measured using the GDS-15. The findings suggest that lower levels of serum 25-OH-D are associated with depressive symptoms in a population of older adults. Moreover, this association remained after adjusting for potential confounding factors related to sociodemographic, health, and lifestyle variables. These results may have important implications regarding the provision of novel therapeutic targets for symptoms of depression in a population of older adults.

In addition, these results are in line with published meta-analyses and reviews on vitamin D status and symptoms of depression [13]. Studies have found that low vitamin D at baseline is associated with depressive symptoms in older adults [8,17,23]. Several direct and indirect biological mechanisms have been hypothesized to describe these associations, including the role of vitamin D in regulating neurotransmitters, dopamine, acetylcholine, and noradrenaline as well as in affecting neurotrophic factors [47]. Vitamin D receptors and the vitamin D-activating enzyme, 1a-hydroxylase, are present in regions of the human brain (for example, the prefrontal cortex and hippocampus), which are recognized as being associated with the pathophysiology of depression [48,49]. Vitamin D also helps regulate serotonin production and may indirectly fight depressive symptoms via its proposed anti-inflammatory effect [3,18,21]. It has been indicated that one action of vitamin D is to stimulate the expression of the serotonin-synthesizing gene, tryptophan hydroxylase 2, while suppressing the expression of tryptophan hydroxylase 1. Both tryptophan hydroxylase 1 and tryptophan hydroxylase 2 play vital roles in serotonin synthesis [18]. Thus, vitamin D may prevent depression by preserving normal serotonin levels [18,50]. However, despite the beneficial connection between vitamin D and depression, some of the previous literature has failed to confirm this association [23,26,51], which is probably due to differences in study design, the various methods used for diagnosing depression, the machinery used to analyze vitamin D, and ethnic/racial diversity.

In this study, the association between serum vitamin D levels and depressive symptoms, observed in a sample comprising both men and women, was stronger in women than in men, which could indicate this group’s greater vulnerability to depression. Previous studies have described gender differences that frequently indicate lower 25-(OH)-D levels in women than in men. In particular, older women with vitamin D deficiency have reported more depressive symptoms [52]. Other studies have likewise revealed a higher prevalence of depressive symptoms among women, such as, for example, studies conducted in China [4], South Korea [26], England [51], and southern Brazil [20]. Moreover, in Kuwait, symptoms of depression also appear to be more common among women [53]. The reasons for these differences between men and women are likely to be that the major determinants of depression differ between them. For example, it is proposed that, in men, depression is mainly influenced by physical factors, whereas in women, depression is mainly influenced by social–psychological factors [54,55]. An examination of vitamin D levels in men and women with similar levels of depression has demonstrated that the levels in men appear to be lower than those measured in women, perhaps indicating that vitamin D levels need to fall further from a normal range before men experience depression, although this needs further investigation. Future research on the effects of vitamin D should examine the differences between men and women in depth when assessing the risk of depressive symptoms among older adults.

Meanwhile, the present results revealed a correlation between serum 25-OH-D concentrations and depressive symptoms in relation to physical activity in a population of older adults. Likewise, a recent meta-analysis revealed that physical activity can confer protection against the occurrence of depressive symptoms [56]. A recent study showed that older adults are not expected to be capable of engaging in a high rate of vigorous physical exercise, reporting that, in order to gain benefits from physical exercise, moderate physical activity is a more realistic approach for older adults [57]. Physical activity, if performed outdoors, is important for both the dermal production of vitamin D and its own potential to relieve depressive symptoms [58]. Conversely, individuals with depression are more likely to stay indoors and have poorer dietary patterns, thereby affecting their levels of vitamin D [59,60].

The current findings show that short sleep duration and low vitamin D levels are both associated with depressive symptoms in a population of older adults, thereby supporting the findings of previous studies. For example, Kim et al. [61] investigated the association between self-reported sleep duration and serum vitamin D levels in older Korean adults. The results suggested that a shorter sleep duration was correlated with lower vitamin D status among older adults. Another study found a relationship between sleep apnea and vitamin D deficiency [62]. The association between both vitamin D deficiency and sleep disorders in an older population may be explained as follows: several studies have identified vitamin D receptors in nearly all tissues in the body, including both neuronal and glial cells in the central nervous system [63]. Vitamin D receptors are present in an area of the brain that regulates the sleep–wake cycle, among other behaviors [50,64,65]. The evidence suggests that sleep disorders and circadian rhythm deregulation are also more common among older people, compared with among younger adults [66]. However, aging is one of the main risk factors for sleep curtailment, which is consequently more common among older adults [64]. Therefore, the shorter sleep duration in older adults may explain, at least in part, the age-related changes in vitamin D levels and metabolism. Nevertheless, new studies are necessary to test these hypotheses.

The present study found no association between vitamin D status and depressive symptoms in relation to sun exposure. In contrast, Cui et al. [65] revealed that increased duration of sunlight exposure was correlated with decreased depressive symptoms in older women aged 60 years and over. Lambert et al. [67] suggested that the prevailing amount of sunlight affects the brain’s serotonergic activity. Inadequate serotonin and low brain serotonergic activity have been associated with both seasonal affective disorder [67] and depression [68]. The discrepancy in these findings may be because the present study sample tended to be limited in their sun exposure overall. The reason for this avoidance was likely to be due to the extremely high temperatures that are typical in this geographic region, especially in summer [41,65].

In the present results, no significant association was observed between vitamin D, PTH, and the presence of severe depressive symptoms in a population of older adults. Two previous studies have produced similar results [69], with Pan et al. [69] and Zhao et al. [70] not finding any relationship between serum 25-(OH)-D, PTH, and depressive symptoms. Conversely, the present results conflict with the findings of Hoogendijk et al. [25], who reported that decreased vitamin D and increased PTH were significantly associated with a high depression score. Studies have suggested that PTH may play a role in the pathogenesis of depression, and vitamin D status could be a direct or indirect intermediating factor. For example, low vitamin D and increased PTH possibly increase inflammation in the body, which is a risk factor for depressive symptoms [71]. These contradictory findings may be due to the different research methods and populations used and to the self-reported depression scales used to assess symptoms of depression (GDS-15 in the present study and the Center for Epidemiologic Studies Depression Scale in Hoogendijk et al. [25]).

Given the high prevalence of depressive symptoms and low serum 25-OH-D levels, a correlation between these two conditions has significant public health implications. The prevention of vitamin D deficiency may become a future strategy to prevent the development of depression [72]. Numerous cost-effective intervention strategies, including vitamin D supplementation, sunlight exposure, and fortified food intake, have been identified as effective ways of improving low vitamin D levels, especially since supplementation with vitamin D is cost-effective and does not have significant adverse effects [4,73]. Furthermore, the increase in the natural production of serotonin brought about by vitamin D could potentially enhance the effect of serotonin taken orally as an antidepressant [18].

Some studies have found an association between vitamin D supplementation and depression [46,74], while others have found no association [75,76]. In the present study, only 42.2% of the participants were taking vitamin D supplements, despite depression and vitamin D deficiency being common conditions in older adults [74]. Moreover, the majority of those taking vitamin D supplements were in the group with a normal level of depressive symptoms, which declined with an increase in the severity of depression, where 0% of those with severe depressive symptoms were taking such supplements. The univariate analysis uncovered a significant association between the GDS-15 score and average 25-(OH)-D level in the participants receiving vitamin D supplements. In the present study, the effects of vitamin D supplementation on depressive symptoms were not specifically examined, but information on vitamin D and calcium supplements was used because these can affect vitamin D levels. Future studies could further examine these relationships given the inconsistent findings from previous work, particularly that on populations of older adults in the Arabian Gulf. Furthermore, the clinical implications of these results indicate that healthcare experts should be mindful of this relationship in the older adult population so that suitable interventions can be provided.

Strengths and Limitations

This study had several strengths. For instance, this is the first study to measure vitamin D levels and symptoms of depression in a large, nationally representative sample of elderly adults in Kuwait or, indeed, anywhere in the Gulf region. Samples were collected from senior citizens from Kuwait’s six Governorates. This meant that the sample was representative of older adults in Kuwait, both male and female and aged 65 years and above. Additionally, this study considered numerous covariates that are important in the relationship between vitamin D status and depressive symptoms, such as skin color and the season in which blood samples are collected. Additionally, the 15-item GDS, which evaluates depressive symptoms, has been widely used as a screening tool in similar studies and validated for application in the Kuwaiti population.

Validated and standardized questionnaires were also administered to obtain data on sun exposure, physical activity, food frequency, and general use of vitamin D supplementation. However, with all self-reported information (e.g., IPAQ and SFFQ), there is the potential for reporting bias, with respondents tending to over-report socially desirable behaviors and to under-report behaviors that are viewed as less desirable. Specifically, the MMSE may be influenced by language, culture, and education level. Levels of 25(OH)D were measured using MC/MS/MS, which is considered the most sensitive and precise method available for determining 1,25-(OH)2-D2, while D3 serum 25(OH)D concentration reflects the vitamin D obtained from dietary sources, supplements, and subcutaneous synthesis, stimulated by exposure to UVB radiation from sunlight. However, the present study also has some limitations. Due to its cross-sectional design, it was not possible to draw conclusions regarding causal associations. Second, depression was assessed via a self-reported scale and not through clinical diagnosis, although only participants who were older and not on antidepressant medication were included in this study. Lastly, kidney function measurements, which could have impacted on vitamin D status and provided additional insights into our findings, were not included in the statistical analysis.

## 5. Conclusions

In conclusion, it was found that the symptoms and severity of depression, measured using the GDS-15 scale, were strongly inversely associated with serum 25-(OH)-D, even after adjustment for sex, physical activity, health status, and marital status. These findings could have significant public health implications for the identification and treatment of depressive symptoms among older adults. Well-designed RCTs investigating the administration of vitamin D for the prevention and treatment of depression in older adults with a concurrent vitamin D deficiency < 50 nmol/L (20 ng/mL) who are depressed at baseline are key to this area of research. These trials should have dosing protocols, uniform assays, and an adequate control of confounders.

## Figures and Tables

**Figure 1 nutrients-14-01548-f001:**
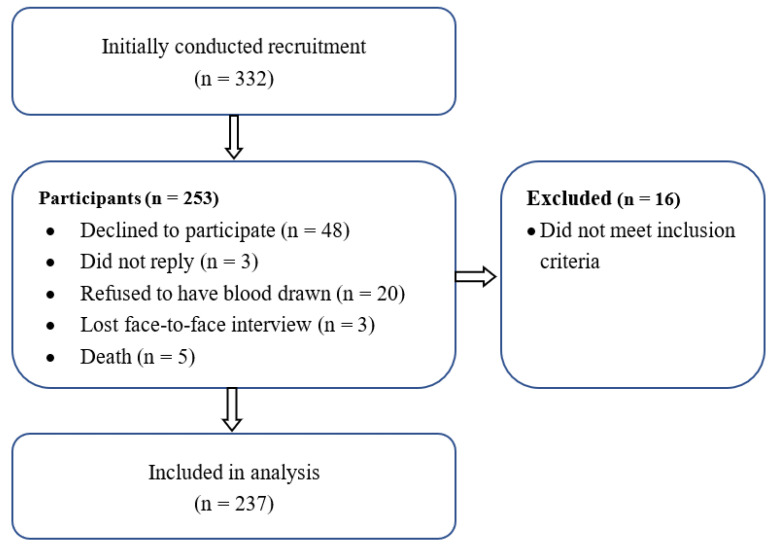
Consort diagram of the cross-sectional study.

**Figure 2 nutrients-14-01548-f002:**
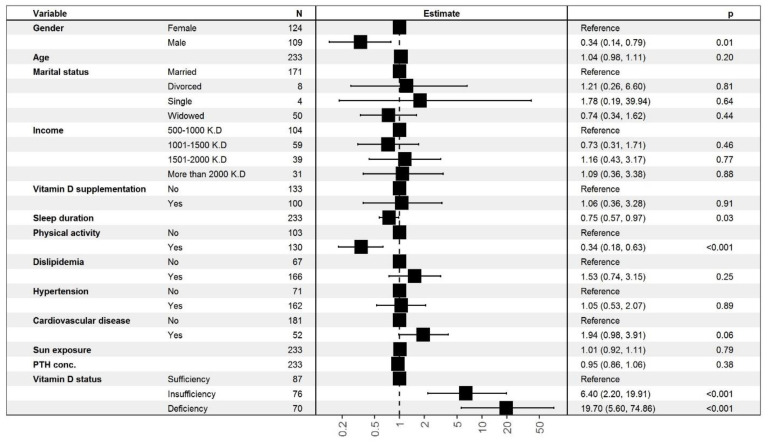
Multivariate association between GDS-15 and 25(OH)D.

**Table 1 nutrients-14-01548-t001:** Social demographic factors associated with symptoms of depression (GDS-15) among the study participants.

	[ALL]	Normal	Mild	Moderate	Severe	*p*-Value
Variables	*n* = 237	*n* = 34	*n* = 100	*n* = 100	*n* = 3	
Age, mean (SD), year	71.4 (4.94)	70.3 (4.12)	71.3 (4.58)	71.9 (5.46)	69.5 (6.36)	0.158
Gender *n* %						0.003
Female	128 (54.0%)	16 (47.1%)	43 (43.0%)	66 (66.0%)	3 (100%)	0.001
Male	109 (46.0%)	18 (52.9%)	57 (57.0%)	34 (34.0%)	0 (0.00%)	0.001
Marital status, *n* %						0.042
Divorced	8 (3.38%)	2 (5.88%)	1 (1.00%)	5 (5.00%)	0 (0.00%)	0.072
Married	173 (73.0%)	26 (76.5%)	81 (81.0%)	65 (65.0%)	1 (33.3%)	<0.001
Single	4 (1.69%)	0 (0.00%)	1 (1.00%)	3 (3.00%)	0 (0.00%)	0.112
Widowed	52 (21.9%)	6 (17.6%)	17 (17.0%)	27 (27.0%)	2 (66.7%)	<0.001
Income per month, *n* (%)						0.012
500–1000 K.D	104 (44.6%)	9 (26.5%)	39 (40.2%)	53 (53.5%)	3 (100%)	<0.000
1001–1500 K.D	59 (25.3%)	14 (41.2%)	24 (24.7%)	21 (21.2%)	0 (0.00%)	<0.000
1501–2000 K.D	39 (16.7%)	5 (14.7%)	20 (20.6%)	14 (14.1%)	0 (0.00%)	<0.000
More than 2000 K.D	31 (13.3%)	6 (17.6%)	14 (14.4%)	11 (11.1%)	0 (0.00%)	<0.002
Education level, *n* (%)						0.071
No formal education	51 (21.5%)	9 (26.5%)	15 (15.0%)	26 (26.0%)	1 (33.3%)	0.001
Completed primary/intermediate school	36 (15.2%)	1 (2.94%)	23 (23.0%)	11 (11.0%)	1 (33.3%)	0.001
Completed secondary school	55 (23.2%)	7 (20.6%)	22 (22.0%)	25 (25.0%)	1 (33.3%)	0.004
Completed Diploma	43 (18.1%)	7 (20.6%)	16 (16.0%)	20 (20.0%)	0 (0.00%)	0.050
University degree or above	52 (21.9%)	10 (29.4%)	24 (24.0%)	18 (18.0%)	0 (0.00%)	0.058
Type of housing, *n* (%)						0.170
Rental flat	3 (1.27%)	0 (0.00%)	1 (1.00%)	2 (2.00%)	0 (0.00%)	NA
Rental house	2 (0.84%)	0 (0.00%)	0 (0.00%)	2 (2.00%)	0 (0.00%)	NA
Owned flat	1 (0.42%)	0 (0.00%)	0 (0.00%)	1 (1.00%)	0 (0.00%)	NA
Owned house	231 (97.5%)	34 (100%)	99 (99.0%)	95 (95.0%)	3 (100%)	<0.001
Having children	6.59 (3.07)	6.50 (3.10)	6.81 (2.98)	6.33 (3.14)	8.67 (3.51)	0.757
Sleep duration (day/hours; mean (SD)	6.41 (1.10)	6.82 (1.10)	6.52 (1.00)	6.16 (1.15)	6.50 (2.12)	0.002
Smoking cigarettes (yes/no), *n* (%)	17 (7.17%)	2 (5.88%)	7 (7.00%)	8 (8.00%)	0 (0.00%)	0.772
Drinking alcohol (yes/no), *n* (%)	4 (1.69%)	0 (0.00%)	2 (2.00%)	2 (2.00%)	0 (0.00%)	0.585
Walking per minutes, mean (SD)	3.16 (2.99)	4.1 (2.90)	3.6 (3.00)	2.5 (3.10)	0.00 (0.00)	0.004
Physical activity *n* (%)	108 (46.2%)	9 (28.1%)	37 (37.4%)	59 (59.0%)	3 (100%)	<0.001
Vitamin D supplement (yes/no), *n* (%)	100 (42.2%)	28 (82.4%)	43 (43.0%)	29 (29.0%)	0 (0.00%)	<0.001
Calcium supplement (yes/no), *n* (%)	7 (2.95%)	2 (5.88%)	3 (3.00%)	2 (2.00%)	0 (0.00%)	0.261
Dietary intake of vitamin D (IU) *	247 (261)	307 (277)	245 (242)	233 (277)	114 (76.0)	0.157
Dietary intake of calcium (mg) *	871 (565)	1072 (564)	919 (552)	772 (564)	404 (177)	0.002
Sun Exposure	1.27 (3.03)	2.21 (2.85)	1.42 (3.44)	0.83 (2.62)	0.00 (0.00)	0.014

Continuous variables were expressed as mean ± standard deviation (SD); the categorical variables were expressed as number (*n*) and percentage. (GDS-15), Geriatric Depression Scale-15; KD; Kuwait dinar. * Dietary intake of vitamin D (IU/day per 1000 kcal). * Dietary intake of calcium (mg/day per 1000 kcal). NA = not applicable, due to a very small sample size.

**Table 2 nutrients-14-01548-t002:** Association between the Geriatric Depression symptom scale-15 (GDS-15), anthropometric measurements, and laboratory parameters.

	[ALL]	Normal	Mild	Moderate	Severe	*p*-Value
Variables	*n* = 237	*n =* 34	*n* = 100	*n* = 100	*n* = 3	
Systolic mm Hg; mean (SD)	133 (12.2)	131 (11.1)	132 (11.7)	135 (13.0)	133 (15.3)	0.361
Diastolic mm Hg; mean (SD)	78.7 (11.6)	80.2 (20.8)	77.7 (6.62)	79.2 (11.6)	77.7(4.04)	0.681
Serum 25-OH-D level (nmol/L), *n* (%)						<0.001
Deficiency < 50 nmol/L	72 (30.4%)	1 (2.94%)	27 (27.0%)	42 (42.0%)	2 (66.7%)	<0.001
Insufficiency: 50–75 nmol/L	78 (32.9%)	3 (8.82%)	37 (37.0%)	37 (37.0%)	1 (33.3%)	<0.001
Sufficiency ≥ 75 nmol/L	87 (36.7%)	30 (88.2%)	36 (36.0%)	21 (21.0%)	0 (0.00%)	0.140
Dyslipidemia	170 (71.7%)	18 (52.9%)	72 (72.0%)	77 (77.0%)	3 (100%)	0.008
Hypertension	165 (69.6%)	21 (61.8%)	66 (66.0%)	76 (76.0%)	2 (66.7%)	0.084
Type 2 diabetes	152 (64.1%)	19 (55.9%)	62 (62.0%)	68 (68.0%)	3 (100%)	0.099
Cardiovascular disease	54 (22.9%)	4 (11.8%)	22 (22.0%)	26 (26.3%)	2 (66.7%)	0.037
Osteoporosis (OA)	59 (24.9%)	6 (17.6%)	24 (24.0%)	28 (28.0%)	1 (33.3%)	0.208
Pigmentary phototype:						0.107
II	3 (1.27%)	0 (0.00%)	3 (3.00%)	0 (0.00%)	0 (0.00%)	NA
III	58 (24.5%)	12 (35.3%)	22 (22.0%)	22 (22.0%)	2 (66.7%)	0.118
IV	154 (65.0%)	22 (64.7%)	65 (65.0%)	66 (66.0%)	1 (33.3%)	<0.001
V	22 (9.28%)	0 (0.00%)	10 (10.0%)	12 (12.0%)	0 (0.00%)	0.670
Seasonality, *n* (%)						0.338
Summer/Fall	82 (34.6%)	11 (32.4%)	30 (30.0%)	41 (41.0%)	0 (0.00%)	<0.001
Winter/Spring	155 (65.4%)	23 (67.6%)	70 (70.0%)	59 (59.0%)	3 (100%)	<0.001

Continuous variables were expressed as mean ± standard deviation (SD) and the categorical variables were expressed as number (*n*) and percentage. (GDS-15) Geriatric Depression Scale-15; Pigmentary Phototype: (II) fair, (III) fair to medium, (IV) medium, and (V) olive or dark. SBP, systolic blood pressure; DBP, diastolic blood pressure function. NA = not applicable due to a very small sample.

**Table 3 nutrients-14-01548-t003:** Association between the Geriatric Depression symptom scale-15 (GDS-15), anthropometric measurements, and laboratory parameters.

Variables	Normal	Mild	Moderate	Severe	*p*-Value
	*n* = 34	*n* = 100	*n* = 100	*n* = 3	
BMI (kg/m^2^), *n* (%)					<0.001
Normal weight Overweight Obese	8 (23.5%) 16 (47.1%) 10 (29.4%)	12 (12.2%) 43 (43.9%) 43 (43.9%)	42 (42.0%) 37 (37.0%) 21 (21.0%)	2 (66.7%) 1 (33.3%) 0 (0.00%)	<0.001 <0.01 <0.001
Waist–hip ratio WHR, *n* (%)					<0.001
Low Moderate High	10 (29.4%) 9 (26.5%) 15 (44.1%)	33 (33.0%) 16 (16.0%) 51 (51.0%)	26 (26.3%) 14 (14.1%) 59 (59.6%)	0 (0.00%) 1 (33.3%) 2 (66.7%)	<0.01 0.342 <0.001
Waist (cm) mean (SD)	98.3 (26.0)	103 (17.8)	102 (16.8)	101 (15.3)	0.610
Hip (cm) mean (SD)	102 (30.5)	108 (15.2)	109 (12.7)	111 (9.83)	0.064
25-OH-D nmol/L	100 (31.7)	71.2 (38.6)	58.6 (30.1)	49.0 (6.93)	<0.001
ALP IU/L	72.6 (22.8)	71.0 (19.1)	76.2 (29.0)	102 (40.5)	0.104
PO4 mmol/L	1.13 (0.13)	1.12 (0.17)	1.12 (0.16)	1.43 (0.29)	0.013
Ca mmol/L	2.30 (0.10)	2.30 (0.11)	2.31 (0.17)	2.21 (0.07)	0.600
PTH pmol/L	5.55 (2.17)	5.89 (2.45)	6.19 (3.23)	15.4 (18.6)	<0.001
Cortisol (AM) nmol/L	379 (159)	384 (135)	375 (143)	252 (64.3)	0.464
Glucose (F) mmol/L	6.15 (1.55)	6.74 (2.12)	7.77 (8.66)	12.6 (11.1)	0.169
Insulin ulU/mL	13.4 (6.62)	16.2 (11.6)	15.3 (10.7)	10.2 (7.40)	0.466

Continuous variables were expressed as mean ± standard deviation (SD). Abbreviations: 25-(OH)-D: total serum 25-hydroxyvitamin D; ALP: alkaline phosphatase; Ca: calcium; PTH: parathyroid hormone; PO4: phosphate; (%) BMI: body mass index, calculated as weight in kilograms divided by height in meters squared.

## Data Availability

The data presented in this study are available from the corresponding author upon request.

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
