# Peer review of "Lower Blood Vitamin D Levels Are Associated with Depressive Symptoms in a Population of Older Adults in Kuwait: A Cross-Sectional Study"

_nutrients, 2022, doi:10.3390/nu14081548_

Round 1
Reviewer 1 Report
The present study tried to examine the association between vitamin D deficiency and depression in adults aged 65 years and older from Kuwait. This study is relevant because show data from a population previously not described in the field.
Title
This must be adjusted to the findings of the study and also, include the nationality of population studied.
Introduction
The authors stated that “Depression is a common disorder among the elderly in general, but depressive symptoms are most common in the most senior adults...” (see lines 38 and 39). It is very confusing, because depression is a very well described clinical entity and the fact the depressive symptoms define depression, why the authors describe this as two different conditions? Actually the depressive scale used by the authors is based in depressive symptoms. It is highly recommendable do not make this difference across all manuscript.
This section must be revised, because there is some paragraphs very redundant. For example, see lines 81 to 83, two sentences to describe the same. Please, revise carefully the English wording.
Methods and materials
Each assessment must be described in their subtitle section. For example, see lines 126 to 131, the authors described en general all the measurements realized in the study. It is unnecessary because each protocol was mentioned with certain detail below.
See lines 133 – 134 are not useful, it must be deleted.
Change “ Laboratory Assays” (line 173) by Biochemical assessment
Results
The tables and figures must be included in the appropriated places, according these are cited in the text.
All the legends must be rewritten.
Change “ Laboratory Assays” (line 173) by Biochemical assessment
The authors must include all the P-values in all subcategories shown in the table 1 and 2 to make sure where are observed the real differences.
The results described in figures 2 and 3 must be add as supplemental material, the data showed in those figures did not add critical information to data shown in the Table 1, 2 and figure 4.
Discussion
It is very confusing that the authors stated “...This study showed a clear association between serum 25(OH)D concentration and depressive symptom scores measured using the GDS-15. Our findings suggest that low levels of serum 25(OH)D are independently associated with depressive symptoms in an elderly population...” (see lines 353-355) This state did not reflect the results showed and are contradictory to the data showed in table 1 and figure 4. And below, the authors discussed again their results with others studies where the low levels of vitamin D are associated with depression. I think, the authors must rewrite the discussion in as uniform sense.
The authors did not discuss why the vitamin D supplement are not enough to prevent mild and moderate depression in their study group? The authors did not measure kidney function in the studied subjects to explain why the low vitamin D levels and how the nutritional status could affect the benefits of vitamin d supplementation?
Author Response
Reviewer (1) Comments |
Our Response |
The present study tried to examine the association between vitamin D deficiency and depression in adults aged 65 years and older from Kuwait. This study is relevant because show data from a population previously not described in the field.
|
Thank you for taking the time to review our manuscript, your comments have helped us strengthen our work significantly. All your valuable comments have been noted and the manuscript thoroughly reviewed, accordingly. |
Title This must be adjusted to the findings of the study and also, include the nationality of population studied.
|
Thank you; the title has been adjusted, accordingly, to include the nationality of the population studied. The new title is: Lower Serum Vitamin D Levels Are Associated with Depressive Symptoms in an Elderly Population in Kuwait: A Cross-sectional Study |
Introduction The authors stated that “Depression is a common disorder among the elderly in general, but depressive symptoms are most common in the most senior adults...” (see lines 38 and 39). It is very confusing, because depression is a very well described clinical entity and the fact the depressive symptoms define depression, why the authors describe this as two different conditions? Actually, the depressive scale used by the authors is based in depressive symptoms. It is highly recommendable do not make this difference across all manuscript.
|
We have revised the introduction and the rest of the paper to not make a distinct difference between depression and depressive symptoms.
|
This section must be revised, because there is some paragraphs very redundant. For example, see lines 81 to 83, two sentences to describe the same. Please, revise carefully the English wording.
|
Thank you; We have revised this section to remove the redundancy mentioned and to address any English-language issues.
|
Methods and materials Each assessment must be described in their subtitle section. For example, see lines 126 to 131, the authors described en general all the measurements realized in the study. It is unnecessary because each protocol was mentioned with certain detail below. See lines 133 – 134 are not useful, it must be deleted.
|
We have removed the mention of general measurements from this section and limited the section exclusively to presenting the primary outcome of the depression levels.
Thank you; We have removed these lines.
|
Change “Laboratory Assays” (line 173) by Biochemical assessment
|
We have changed the Laboratory Assays” (line 173) to Biochemical Assessment |
Results The tables and figures must be included in the appropriated places, according these are cited in the text. All the legends must be rewritten.
|
We have placed all the figures and tables close to where they are first mentioned in the text. We have ensured all figures and tables are mentioned within the text. It is unclear why the reviewer would like the legends to be rewritten. We have reviewed the legends and made changes to enhance clarity. |
Change “Laboratory Assays” (line 173) by Biochemical assessment |
We have changed the Laboratory Assays” (line 173) to Biochemical assessment |
The authors must include all the P-values in all subcategories shown in the table 1 and 2 to make sure where are observed the real differences. |
We included the p-value for all subcategories (in Tables 1, 2 and 3) to verify where the real differences were observed. The p-value was calculated using a chi square |
The results described in figures 2 and 3 must be add as supplemental material, the data showed in those figures did not add critical information to data shown in the Table 1, 2 and figure 4. |
We have moved figures 2 and 3 to supplementary material and adjusted the numbering and naming of the figures in the manuscript accordingly
|
Discussion It is very confusing that the authors stated “...This study showed a clear association between serum 25(OH)D concentration and depressive symptom scores measured using the GDS-15. Our findings suggest that low levels of serum 25(OH)D are independently associated with depressive symptoms in an elderly population...” (see lines 353-355) This state did not reflect the results showed and are contradictory to the data showed in table 1 and figure 4. And below, the authors discussed again their results with others studies where the low levels of vitamin D are associated with depression. I think, the authors must rewrite the discussion in as uniform sense.
|
We have toned down the language of this sentence to describe the results however, we believe this summary of the results reflects the data presented in the paper. The data in figure 4 (now figure 2) shows that low vitamin D status (insufficiency and deficiency) are associated with a 6.4 to 19.7 fold higher odds of a higher depression score. In table 1 it is shown that vitamin D deficiency is much more common among those with any level of depressive symptoms (2.94% with deficiency at the “normal” level vs. >27% deficiency at any other level). |
The authors did not discuss why the vitamin D supplement are not enough to prevent mild and moderate depression in their study group? The authors did not measure kidney function in the studied subjects to explain why the low vitamin D levels and how the nutritional status could affect the benefits of vitamin d supplementation? |
We have added a point to the discussion section about the use of vitamin D supplementation and its lack of use among those with more severe depressive symptoms. We also referenced the existing literature which demonstrates that the relationship between vitamin D supplementation and depressive symptoms is not consistent. We did measure kidney function and have updated the methods to reflect this. Screening was performed to ensure that kidney function was normal in all the participants. The reason for this was because certain diseases can interfere with vitamin D synthesis and metabolism. An example of kidney function screening would involve the measurement of Na, K, Cl, creatinine, albumin, urea, uric acid, and total bilirubin. Other screening blood tests were conducted for liver function (gamma GT level, ALT, and ALP) and a complete blood count (CBC). The above-mentioned blood parameters were required to fall within the normal range. However, in this study, the kidney function test results were not included in the statistical analysis conducted; instead, they were merely for screening purposes. |
References
- Chaaya M, Sibai AM, Roueiheb ZE, Chemaitelly H, Chahine LM, Al-Amin H, et al. Validation of the Arabic version of the short Geriatric Depression Scale (GDS-15). International psychogeriatrics. 2008;20(3):571-81.
- Brown LM, Schinka JA. Development and initial validation of a 15‐item informant version of the Geriatric Depression Scale. International Journal of Geriatric Psychiatry: A journal of the psychiatry of late life and allied sciences. 2005;20(10):911-8.
- Papandreou D, Rachaniotis N, Lari M, Al Mussabi W. Validation of a food frequency questionnaire for vitamin D and calcium intake in healthy female college students. Food and Nutrition Sciences. 2014;5(21):2048.
- Zareef TA, Jackson RT, Alkahtani AA. Vitamin D Intake among Premenopausal Women Living in Jeddah: Food Sources and Relationship to Demographic Factors and Bone Health. Journal of Nutrition and Metabolism. 2018;2018:8570986.
- Helou K, El Helou N, Mahfouz M, Mahfouz Y, Salameh P, Harmouche-Karaki M. Validity and reliability of an adapted arabic version of the long international physical activity questionnaire. BMC public health. 2018;18(1):1-8.
- Toffanello ED, Sergi G, Veronese N, Perissinotto E, Zambon S, Coin A, et al. Serum 25-hydroxyvitamin d and the onset of late-life depressive mood in older men and women: the Pro. VA study. Journals of Gerontology Series A: Biomedical Sciences and Medical Sciences. 2014;69(12):1554-61.
- Albanna M, Yehya A, Khairi A, Dafeeah E, Elhadi A, Rezgui L, et al. Validation and cultural adaptation of the Arabic versions of the Mini-Mental Status Examination - 2 and Mini-Cog test. Neuropsychiatr Dis Treat. 2017;13:793-801.
Reviewer 2 Report
I thank for the opportunity to read the manuscript by Thurayya Albolushi et al. which analyzed the association between vitamin D deficiency and depression in adults aged 65 years and older from Kuwait. The paper is interesting, the authors are aware of the strenght of the study but also the study limitations. Overall, the study is well organized and rather easy to follow. However, English language needs improvement. I have also several comments:
1. Instead of "blood sugar" use "blood glucose".
2. Overall, the manuscript needs formatting.
3. All tables and figures should be placed close to the text where they are first mentioned.
4. I suggest to divide table 1 into two tables: the first one showing demographic variables and the second one presenting clinical variables. In the present form, the Table 1 is too large.
5. Reducing of cited references should be considered.
6. Description of post-hoc pairwise comparisons should be included into the section with statistical analysis.
7. Please specify when the blood pressure measurement was performed (early in the morning, in the afternoon, in the evening?).
Author Response
Reviewer (2) Comments I thank for the opportunity to read the manuscript by Thurayya Albolushi et al. which analyzed the association between vitamin D deficiency and depression in adults aged 65 years and older from Kuwait. The paper is interesting, the authors are aware of the strenght of the study but also the study limitations. Overall, the study is well organized and rather easy to follow. However, English language needs improvement. I have also several comments: |
Thank you for taking the time to review our manuscript, your comments have helped us strengthen our work significantly. All your valuable comments have been noted and the manuscript thoroughly reviewed, accordingly. The English language has been improved
|
1. Instead of "blood sugar" use "blood glucose".
|
Thank you for this comment, we have changed "blood sugar" We used "blood glucose". |
2. Overall, the manuscript needs formatting
|
We have formatted the manuscript according to the Nutrients template. Specifically we have made adjustments to the presentation of the figures and tables to be in line with the text, adjusted paragraph spacing, and consistently bolded/italicized headings and subheadings. |
3. All tables and figures should be placed close to the text where they are first mentioned.
|
We have edited the manuscript; all tables and figures have been placed close to the text where they are first mentioned |
I suggest to divide table 1 into two tables: the first one showing demographic variables and the second one presenting clinical variables. In the present form, the Table 1 is too large.
|
Thank you; these have now been addressed. The table1 has been divided into two tables: the first one showing demographic variables and the second one presenting clinical variable and renumbered all the tables in the manuscript accordingly |
5. Reducing of cited references should be considered. |
Thank you; these have now been addressed. We have removed some of references |
6. Description of post-hoc pairwise comparisons should be included into the section with statistical analysis. |
Thank you; We have included mention of post-hoc pairwise comparisons in the statistical analysis section. Post-hoc pairwise comparisons of the levels of depressive symptoms and vitamin D status were also conducted |
7. Please specify when the blood pressure measurement was performed (early in the morning, in the afternoon, in the evening?). |
We have clarified the methods by which blood pressure was taken in the methods section. Briefly, measurements were performed in the early morning prior to eating or taking any medications. Three separate blood pressure readings were taken and averaged. The Physicians in the geriatric clinic followed the International Society of Hypertension Global Hypertension Practice Guidelines. In addition, we had access to medical history/records(electronic records provided accurate and up-to-date information about each elderly patient), and all participants were referred by medical practitioners who physically examined them to confirm the hypertension diagnosis |
References
- Chaaya M, Sibai AM, Roueiheb ZE, Chemaitelly H, Chahine LM, Al-Amin H, et al. Validation of the Arabic version of the short Geriatric Depression Scale (GDS-15). International psychogeriatrics. 2008;20(3):571-81.
- Brown LM, Schinka JA. Development and initial validation of a 15‐item informant version of the Geriatric Depression Scale. International Journal of Geriatric Psychiatry: A journal of the psychiatry of late life and allied sciences. 2005;20(10):911-8.
- Papandreou D, Rachaniotis N, Lari M, Al Mussabi W. Validation of a food frequency questionnaire for vitamin D and calcium intake in healthy female college students. Food and Nutrition Sciences. 2014;5(21):2048.
- Zareef TA, Jackson RT, Alkahtani AA. Vitamin D Intake among Premenopausal Women Living in Jeddah: Food Sources and Relationship to Demographic Factors and Bone Health. Journal of Nutrition and Metabolism. 2018;2018:8570986.
- Helou K, El Helou N, Mahfouz M, Mahfouz Y, Salameh P, Harmouche-Karaki M. Validity and reliability of an adapted arabic version of the long international physical activity questionnaire. BMC public health. 2018;18(1):1-8.
- Toffanello ED, Sergi G, Veronese N, Perissinotto E, Zambon S, Coin A, et al. Serum 25-hydroxyvitamin d and the onset of late-life depressive mood in older men and women: the Pro. VA study. Journals of Gerontology Series A: Biomedical Sciences and Medical Sciences. 2014;69(12):1554-61.
- Albanna M, Yehya A, Khairi A, Dafeeah E, Elhadi A, Rezgui L, et al. Validation and cultural adaptation of the Arabic versions of the Mini-Mental Status Examination - 2 and Mini-Cog test. Neuropsychiatr Dis Treat. 2017;13:793-801.
Reviewer 3 Report
The manuscript entitled “Low blood vitamin D levels are associated with Depressive Symptoms in an Elderly Population: A Cross-sectional Study” presents interesting issue, however some corrections are needed
- Authors should follow the Instructions for authors while preparing their manuscript.
- ‘serum vitamin D (25-OH-D) concentrations’ – please add the references for cut off
- In this section Authors presented the information associated with role and deficiency of vitamin D deficiency.
- This section should be briefly presented – what do we know and what is the background for this study. Some detailed information about other studies are necessary. The good background should present the history of problem, the current knowledge and scientific "gap", and then authors should present how their study could fill this gap to justify the study.
- ‘The potential association between depression and vitamin D deficiency is supported by similar findings in systematic reviews and meta-analyses of observational studies [17, 18]” – authors presented quite old systematic reviews and meta-analyses. Please update it (e.g. https://www.mdpi.com/2077-0383/10/21/5156; https://pubmed.ncbi.nlm.nih.gov/33809478/; https://www.nature.com/articles/s41598-021-87432-3; https://annals-general-psychiatry.biomedcentral.com/articles/10.1186/s12991-020-00288-1).
- Lines 95-96 – ‘specialized geriatric clinics 95 across Kuwait.’ – from all of them in Kuwait? Please specify the number. (please add the justification for selection, if it is not ). More detailed information about recruitment procedure should be presented.
- ‘Geriatric Depression Scale-15 (GDS-15)’ – What is the original language of the questionnaire? How did the repsondents fill out the questionnaire? Was the questionnaire translated? Who did so? Any validation of the translated questionnaire?
- ‘Geriatric Depression Scale-15 (GDS-15)’ – Was the questionnaire previously validated? What was the accuracy and consistency of this questionnaire?
- More information is needed about the validity and reliability of each measure. Additionally, any limitations in reliability and validity need to be addressed in the discussion.
- ‘Physical Activity Scale for the Elderly (PASE), International Physical Activity Questionnaire for the Elderly (IPAQ)’ – ‘ In its officially validated, Arabic, short-version format.” – please provide the reference.
- Physical Activity Scale for the Elderly (PASE), International Physical Activity Questionnaire for the Elderly (IPAQ)’ – More information is needed about the validity and reliability of each measure. Additionally, any limitations in reliability and validity need to be addressed in the discussion.
- ‘Blood pressure was measured on the left upper arm, using a mercury sphygmomanometer.’ – regardless the fact that mercury sphygmomanometer is as “gold standard”; more information is need. According to „2020 International Society of Hypertension Global Hypertension Practice Guidelines” a single blood pressure measurement does not allow to diagnose hypertension – see: “Usually 2–3 office visits at 1–4-week intervals (depending on the BP level) are required to confirm the diagnosis of hypertension.” (https://www.ahajournals.org/doi/10.1161/HYPERTENSIONAHA.120.15026).
- Mini-Mental State Examination (MMSE) - More information is needed about the validity and reliability of each measure. Additionally, any limitations in reliability and validity need to be addressed in the discussion.
- ‘mean and standard deviation’ - Was the normality of distribution tested? The information about it should be added and authors should be consequent. If data have normal distribution, they should be treated as such, if not, nonparametric tests should be applied. Please specify it.
- Figure 2. – the number on each part of flowchart is unclear?
- Author Contributions – I have doubt about if co-authors have made substantial contributions to conception and design, or acquisition of data, or analysis and interpretation of data.
Minor comments:
- ‘17.90%[5]Many adverse’; ‘mood[7] , and among these,’; IPAQ) [38] . In its – there are some typos
- Lines 267; 317 – some unnecessary bolding
Author Response
Reviewer (3) Comments The manuscript entitled “Low blood vitamin D levels are associated with Depressive Symptoms in an Elderly Population: A Cross-sectional Study” presents interesting issue, however some corrections are needed |
Thank you for taking the time to review our manuscript, your comments have helped us strengthen our work significantly. All your valuable comments have been noted and the manuscript thoroughly reviewed, accordingly. |
Authors should follow the Instructions for authors while preparing their manuscript.
|
We have formatted the manuscript according to the Nutrients template. Specifically, we have made adjustments to the presentation of the figures and tables to be in line with the text, adjusted paragraph spacing, and consistently bolded/italicized headings and subheadings. |
serum vitamin D (25-OH-D) concentrations’ – please add the references for cut off . |
We have included a citation for the Endocrine society to reference these cut-off values used within the manuscript text, specifically at first mention. |
In this section Authors presented the information associated with role and deficiency of vitamin D deficiency. This section should be briefly presented – what do we know and what is the background for this study. Some detailed information about other studies are necessary. The good background should present the history of problem, the current knowledge and scientific "gap", and then authors should present how their study could fill this gap to justify the study. |
We have made alterations to the introduction so that it is much more breif and more closely follows the suggested background content.
|
‘The potential association between depression and vitamin D deficiency is supported by similar findings in systematic reviews and meta-analyses of observational studies [17, 18]” – authors presented quite old systematic reviews and meta-analyses. Please update it (e.g. https://www.mdpi.com/2077-0383/10/21/5156; https://pubmed.ncbi.nlm.nih.gov/33809478/; https://www.nature.com/articles/s41598-021-87432-3; https://annals-general-psychiatry.biomedcentral.com/articles/10.1186/s12991-020-00288-1). Okay Done |
Thank you for the suggested references. We have added the references that investigate an adult population to the introduction where they fit (depending on if their conclusions support the association between vitamin D and depression or not).
|
Lines 95-96 – ‘specialized geriatric clinics 95 across Kuwait.’ – from all of them in Kuwait? Please specify the number. (please add the justification for selection, if it is not ). More detailed information about recruitment procedure should be presented.
|
We have added additional information about the selection of clinics to the methods section. Specifially, there are 18 geriatric clinics, in Kuwait and seven were used in this study. The seven healthcare centers were selected via stratified random sampling from the six governates in Kuwait. The seven centers were selected as follows: one in the Capital, Ha-waly, Jahra, Mubarak Al-Kabeer, and Ahmadi and two in Farwanya. |
‘Geriatric Depression Scale-15 (GDS-15)’ – What is the original language of the questionnaire? How did the repsondents fill out the questionnaire? Was the questionnaire translated? Who did so? Any validation of the translated questionnaire?
|
Thank you, we have clarified these questions in the methods section. Briefly, the original questionnaire (GDS-15) was in English and translated to Arabic by a professional translator and two bilingual psychiatrists. It was also validated [1]. A researcher verbally asked participants each question on the GDS-15 and recorded their response. The GDS-15 was proven to be a good instrument for assessing depression in older adults, without dementia, in community settings and primary care patients. Because it is short and easy to administer, it is recommended that it be used as a routine screening test to identify depression among older adults in primary care settings, and to use it in large population health surveys. |
‘Geriatric Depression Scale-15 (GDS-15)’ – Was the questionnaire previously validated? What was the accuracy and consistency of this questionnaire?
|
Thank you, Yes, the questionnaire was previously validated for its Arabic version by Chaaya et al.[1]. The original English version of GDS-15 has also been validated and was found to have sufficient internal consistency reliability (alpha = 0.86) and retest reliability (r = 0.81) to support its use as a clinical instrument [2]. previously the Arabic version was then piloted within a small group of older adults (n = 10) to ensure that all the terms were derstandable to them. Ambiguities were adjusted accordingly. We have briefly mentioned this in the methods as “…has been validated for use as a clinical instrument”. |
More information is needed about the validity and reliability of each measure. Additionally, any limitations in reliability and validity need to be addressed in the discussion.
|
Please see specific responses above and below these have now been addressed. In addition, Information about vitamin D and calcium supplementation was collected using the Food Frequency Questionnaire (SFFQ), previously validated. The FFQ shows promising validation evidence to be used in the future for assessing vitamin D and calcium intakes [3, 4]. the original questionnaire FFQ was in English and translated to Arabic.The questioner showed Positive correlations were observed of daily vitamin D (r = 0.82, p < 0.001) and calcium intake (r = 0.74, p < 0.001) |
‘Physical Activity Scale for the Elderly (PASE), International Physical Activity Questionnaire for the Elderly (IPAQ)’ – ‘ In its officially validated, Arabic, short-version format.” – please provide the reference. |
Thank you, yes the Questionnaire (IPAQ), its officially validated in Arabic short-version format [5] .previously ,the questionnaires were translated and adapted to Arabic language, and then subjected to back-translation to check the accuracy and consistency of this questionnaire. We have added in the reference[5] |
Physical Activity Scale for the Elderly (PASE), International Physical Activity Questionnaire for the Elderly (IPAQ)’ – More information is needed about the validity and reliability of each measure. Additionally, any limitations in reliability and validity need to be addressed in the discussion. |
The PASE (Arabic version) has been previously validated and the results were promising with good internal consistency (Cronbach’s alpha 0.70–0.75), and excellent reliability of the components (ICC2,1 0.90–0.98). The IPAQ (Arabic version) showed acceptable validity and reliability for assessment of physical activity in a sample of Lebanese adults. We have added in some of this information to the methods section. We have included a statement in the limitations section about the potential for mis-reporting. |
Blood pressure was measured on the left upper arm, using a mercury sphygmomanometer.’ – regardless the fact that mercury sphygmomanometer is as “gold standard”; more information is need. According to „2020 International Society of Hypertension Global Hypertension Practice Guidelines” a single blood pressure measurement does not allow to diagnose hypertension – see: “Usually 2–3 office visits at 1–4-week intervals (depending on the BP level) are required to confirm the diagnosis of hypertension.” (https://www.ahajournals.org/doi/10.1161/HYPERTENSIONAHA.120.15026). |
We have clarified the methods by which blood pressure was taken in the methods section. Briefly, measurements were performed in the early morning prior to eating or taking any medications. Three separate blood pressure readings were taken and averaged. The Physicians in the geriatric clinic followed the International Society of Hypertension Global Hypertension Practice Guidelines. In addition, we had access to medical history/records(electronic records provided accurate and up-to-date information about each elderly patient), and all participants were referred by medical practitioners who physically examined them to confirm the hypertension diagnosis |
Mini-Mental State Examination (MMSE) - More information is needed about the validity and reliability of each measure. Additionally, any limitations in reliability and validity need to be addressed in the discussion.
|
Thank you, the Mini-Mental State Examination (MMSE) is the most common utilized tool in cognition screening tests. In present study, all recruited participants were undergo the Mini-Mental State Examination (MMSE)cognitive function was assessed by administering the 30-item Mini-Mental State Examination (MMSE), whereby scores lower than 24 would indicate cognitive impairment [6]. The examination takes ~7–10 minutes to complete. The total score is 30, and it tests a broad range of cognitive functions, including orientation, recall, attention, calculation, language processing and constructional praxiThe MMSE scale has been validated in an Arabic version [7]. It consists of an 11-item measure cognitive functions: orientation, registration, attention, calculation, recall and language)[7].The final version that was approved by the committee was then back translated to English by a new independent professional translator and was compared to the original version by the committee of psychiatrists. The questionnaires were translated and adapted to Arabic language, and then subjected to back-translation to check the accuracy and consistency of this questionnaire. A pilot study was then carried out to test the language and clarity of the scales in a sample of Arab elderly[7]. We have included a statement in the limitations section about the potential for misreporting. |
‘mean and standard deviation’ - Was the normality of distribution tested? The information about it should be added and authors should be consequent. If data have normal distribution, they should be treated as such, if not, nonparametric tests should be applied. Please specify it. |
The data in the present study were treated as normally to the central limit theorum the data can be treated as following a normal distribution ( due to large sample size used in the study n= 237). We have added to the methods that the data was treated as normally distributed. |
Figure 2. – the number on each part of flowchart is unclear?
|
We have replaced figure 2 (now supplementary figure 1 ) with a clearer version
|
Author Contributions – I have doubt about if co-authors have made substantial contributions to conception and design, or acquisition of data, or analysis and interpretation of data.
|
Thank you; we have adjusted the Author Contributions section to read: ‘All co-authors (Jeremy Spencer and Manal Bouhaimed) contributed to the study design and conception. The day-to-day implementation and delivery of the trial was conducted by Thurayya Albolushi. All authors contributed to the drafting of the manuscript and read and approved the final version of this manuscript.’
|
Minor comments: ‘17.90%[5]Many adverse’; ‘mood[7] , and among these,’; IPAQ) [38] . In its – there are some typos Lines 267; 317 – some unnecessary bolding
|
these have now been addressed. We have fixed these errors and removed the extra spaces |
References
- Chaaya M, Sibai AM, Roueiheb ZE, Chemaitelly H, Chahine LM, Al-Amin H, et al. Validation of the Arabic version of the short Geriatric Depression Scale (GDS-15). International psychogeriatrics. 2008;20(3):571-81.
- Brown LM, Schinka JA. Development and initial validation of a 15‐item informant version of the Geriatric Depression Scale. International Journal of Geriatric Psychiatry: A journal of the psychiatry of late life and allied sciences. 2005;20(10):911-8.
- Papandreou D, Rachaniotis N, Lari M, Al Mussabi W. Validation of a food frequency questionnaire for vitamin D and calcium intake in healthy female college students. Food and Nutrition Sciences. 2014;5(21):2048.
- Zareef TA, Jackson RT, Alkahtani AA. Vitamin D Intake among Premenopausal Women Living in Jeddah: Food Sources and Relationship to Demographic Factors and Bone Health. Journal of Nutrition and Metabolism. 2018;2018:8570986.
- Helou K, El Helou N, Mahfouz M, Mahfouz Y, Salameh P, Harmouche-Karaki M. Validity and reliability of an adapted arabic version of the long international physical activity questionnaire. BMC public health. 2018;18(1):1-8.
- Toffanello ED, Sergi G, Veronese N, Perissinotto E, Zambon S, Coin A, et al. Serum 25-hydroxyvitamin d and the onset of late-life depressive mood in older men and women: the Pro. VA study. Journals of Gerontology Series A: Biomedical Sciences and Medical Sciences. 2014;69(12):1554-61.
- Albanna M, Yehya A, Khairi A, Dafeeah E, Elhadi A, Rezgui L, et al. Validation and cultural adaptation of the Arabic versions of the Mini-Mental Status Examination - 2 and Mini-Cog test. Neuropsychiatr Dis Treat. 2017;13:793-801.
Round 2
Reviewer 2 Report
The manuscript was improved. I have no further comments.
Author Response
thank you for taking the time to review our manuscript, your comments have helped us to strengthen our work significantly. the English language has been improved. Editing by English services
Reviewer 3 Report
The manuscript has been improved according to the comments and suggestions. However, the recommendations of the Nutrients (authors guideline) must be followed. There are still plenty of typos!!!
It should be ‘All authors (Thurayya Albolushi, Manal Bouhaimed and Jeremey Spencer) contributed to the study design and conception...’ instead of ‘All co-authors (Jeremy Spencer and Manal Bouhaimed) contributed to the study design and conception...’.
Line 497 - it should be ‘25(OH)D’ instead of ‘d 25-()OH-D’
Line 500 - it should be ‘25(OH)D’ instead of ‘25-OH-D’
Line 464 – it should be ‘1,25-(OH)2-D2’ instead of 1.25-(OH)2-D2’
The nomenclature associated with the name od vitamin D (e.g.) – should be verified!
Author Response
Reviewer (3) Comments The manuscript has been improved according to the comments and suggestions. However, the recommendations of the Nutrients (authors guideline) must be followed. There are still plenty of typos!!!
|
Thank you for taking the time to review our manuscript, your comments have helped us strengthen our work significantly. All your valuable comments have been noted and the manuscript thoroughly reviewed, accordingly. The English language has been improved. Editing by English services |
1-It should be ‘All authors (Thurayya Albolushi, Manal Bouhaimed and Jeremey Spencer) contributed to the study design and conception...’ instead of ‘All co-authors (Jeremy Spencer and Manal Bouhaimed) contributed to the study design and conception...’. |
These have now been addressed. Author Contributions: T.A., M.B., and J.S. contributed to the study design, conception, analysis, and interpretation of the data. The day-to-day implementation of the trial was conducted by T.A. All authors have read and agreed to the published version of the manuscript. |
2-Line 497 - it should be ‘25(OH)D’ instead of ‘d 25-()OH-D’
|
Thank you, these have now been addressed. We have fixed this error and changed it to 25(OH)D |
3- Line 500 - it should be ‘25(OH)D’ instead of ‘25-OH-D’
|
these have now been addressed. We have changed to 25(OH)D |
4-Line 464 – it should be ‘1,25-(OH)2-D2’ instead of 1.25-(OH)2-D2’
The nomenclature associated with the name od vitamin D (e.g.) – should be verified!
|
Thank you, these have now been addressed. We have fixed this error and changed it to 1,25-(OH)2-D2. All nomenclature associated with the name od vitamin D have been verified. |